# Radiation Synthesis of Selenium Nanoparticles Capped with *β*-Glucan and Its Immunostimulant Activity in Cytoxan-Induced Immunosuppressed Mice

**DOI:** 10.3390/nano11092439

**Published:** 2021-09-18

**Authors:** Nguyen Thi Dung, Tran Duc Trong, Nguyen Thanh Vu, Nguyen Trong Binh, Tran Thi Le Minh, Le Quang Luan

**Affiliations:** 1Biotechnology Center of Ho Chi Minh City, Ho Chi Minh City 700000, Vietnam; thuydung9810@gmail.com (N.T.D.); trong21052011@gmail.com (T.D.T.); ntvu1412@gmail.com (N.T.V.); nguyentrongbinhcnsh@yahoo.com (N.T.B.); 2Faculty of Biologinal Sciences, Nong Lam University, Ho Chi Minh City 700000, Vietnam; ttlminh@hcmuaf.edu.vn

**Keywords:** γ-irradiation, selenium nanoparticles, immunostimulant, immunosuppressed mice, water-soluble *β*-glucan

## Abstract

Selenium nanoparticles (SeNPs) with diameters from 64.8 to 110.1 nm were successfully synthesized by γ-irradiation of solutions containing Se^4+^ and water-soluble yeast *β*-glucan. The size and size distribution of SeNPs were analyzed by dynamic light scattering (DLS). Analytical X-ray diffraction (XRD) pattern results confirmed the crystal structure of the Se nanoparticles and Fourier transform infrared (FTIR) spectroscopy revealed that *β*-glucan could interact with SeNPs through steric (Se…O) linkages leading to a homogeneous and translucent solution state for 60 days without any precipitates. In vivo tests in cytoxan-induced immunosuppressed mice revealed that the daily supplementation of SeNPs/*β-*glucan at concentrations of 6 mg per kg body weight of tested mice significantly stimulated the generation of cellular immune factors (white blood cells, neutrophil, lymphocyte, B cells, CD4+ cells, CD34+ cells and natural killer cells) and humoral immune indexes (IgM, IgG, TNF-α, IFN-γ and IL-2) in peripheral blood, bone marrow and spleen of the immunosuppressed mice. The obtained results indicated that radiation-synthesized SeNPs/*β-*glucan may be a candidate for further evaluation as an agent for the prevention of immunosuppression in chemotherapy.

## 1. Introduction

Selenium (Se) is an trace element essential for maintaining the health of mammalian animals due to its broad functions in biologic systems such as antioxidant, immune modulation, cancer prevention and antiviral activities [1,2,3]. However, the margin between the lowest acceptable intake level and the toxic level of selenium compounds is extremely narrow [4] and the toxicity depends on its chemical form. Selenate (Se^+6^), selenite (Se^+4^) and selenide (Se^−2^) are commonly oxidation states of this metalloid element in the environment. Generally, selenium nanoparticles (SeNPs) are prepared by reduction of higher oxidation states to the Se^0^ form [5,6,7]. SeNPs have attracted widespread attention due to their excellent bioavailability, high bioactivity and low toxicity compared to their inorganic and organic counterparts [7,8,9]. Zang et al. [10], and Zhai et al. [11] reported that the LD_50_ of SeNPs in mice was about 91.2–258.2 mg Se per kg body weight and these values were much lower than that of methylselenocysteine (22.0 mg kg^−1^) and H_2_SeO_3_ (14.6 mg kg^−1^). It is reported that SeNPs have higher antioxidant, immunostimulant, anticancer, antiviral and antibacterial effects compared to their ionic forms [12]. The in vivo immunostimulatory effect of biogenic SeNPs on a 4T1 breast cancer model mice carried out by Yazdi et al. [13,14] demonstrated that cellular immunomodulatory components such as granzyme B, IL-12, IFN-γ, and IL-2 were significantly increased in mice treated with both SeNPs and 4T1 cells in comparison to other groups, while the levels of TGF-*β* were decreased in the tested mice. Jia et al. [15] also reported that biosynthesized SeNPs stabilized in lentinan (a branched β-(1,3)-D-glucan extracted from the fruiting bodies of *Lentinus edodes*) showed higher antitumor activity on HeLa cells than that of a lentinan sample. Recently, Duy et al. [16] reported that SeNPs/oligochitosan synthesized by γ-irradiation could restore the white blood cell levels in irradiated mice. However, to the best of our knowledge, the immunostimulant effect of SeNPs on immunosuppressed mice has not been studied.

So far, SeNPs have been synthesized from ionic selenium solutions by different methods, such as biological [13], chemical [15] and irradiation methods [7,16]. According to Hien et al. [7], γ-ray irradiation could be considered a more effective method than others thanks to several advantages such as, the reaction can be performed at room temperature, high-purity SeNPs can be obtained due to the absence of reductant residues, the SeNP size can be easily controlled by adjusting the dose and dose rate, and large-scale production is possible and finally, they are more suited for the medicinal, surgical and pharmaceutical applications.

On the other hand, the stabilizer used also plays an important role in fabrication of SeNP products. Menon et al. [17]. reported that selenium nanoparticles with mean molecular size measurements ranging from 24 nm to 200 nm can be obtained when conjugated with bovine serum albumin (BSA), L-cysteine, glucose, sucrose, chitosan and sulfated polysaccharides as stabilizer in the redox framework. Several biopolymers including curcumin [18], glutathione [19], protein BSA [20,21], galic acid [22] and folic acid [23] have been used to prepare capped SeNPs to enhance to biocompatibility of this product. Several researchers have used natural polysaccharides for stabilizing SeNP products due to their unique properties including excellent biocompatibility, biodegradability, stability, and non-toxicity [7,11,15,16,22,24,25]. According to Zhang et al. [26] polysaccharides can be considered an appropriate template for the synthesis of selenium nanoparticles compared to phenolics and proteins because proteins are prone to enzymatic degradation and require high temperatures and phenolics are auto-oxidized and become aggregated at the pH of the stomach. Among naturally-originated polysaccharides, branched β-(1,3)-D-glucan isolated from yeast cell walls has been widely applied in the food and pharmaceutical industries [27,28,29]. Since the native yeast *β*-glucan is high molecular weight and water-insoluble properties, the degradation of this product to a lower molecular weight and water-soluble form is more suitable for applications [30,31,32,33], including nanoparticle stabilization.

Therefore, the aim of present study was to apply the γ-ray irradiation method for the synthesis of SeNPs stabilized in water-soluble yeast. The immunostimulant effect of the SeNPs/*β*-glucan product was also investigated in cytoxan-induced immunosuppressed mice for evaluation of its potential application as an immunosuppressive nutraceutical.

## 2. Materials and Methods

### 2.1. Materials

Pure selenium dioxide (SeO_2_) and cytoxan (CTX) were supplied by Sigma- Aldrich (St. Louis, MO, USA). ELISA Kits (Mouse IgM-ab133047; Mouse IgG-ab151276; Mouse IL-2-ab223588, Mouse Interferon gamma-ab100689) and Mouse TNF alpha-ab208348) were products of Abcam (Cambridge, UK). Yeast water-soluble *β*-glucan (1–3,1–6) with Mw ~25 kDa was a gift from Biomaterials and Nano Technology Department, Biotechnology Center of Ho Chi Minh City (Vietnam). Adult BALB/c mice were provided by the Stem Cell Institute (Vietnam National University, Ho Chi Minh City, Vietnam). Other pure-grade chemicals were used, and distilled water was used throughout the experiments.

### 2.2. Synthesis of SeNPs/β-Glucan by Gamma Co-60 Irradiation

Solutions of 40, 60, 80, 100 and 120 ppm Se^4+^ in 2% *β*-glucan were prepared by mixing appropriate volumes from 800 ppm Se^4+^ and 2% *β*-glucan stock solutions. pH of Se^4+^/*β*-glucan solution was adjusted to 5.0, 6.0, 7.0 and 8.0 by 0.5% NH_4_OH solution. Then, 100 mL of the prepared Se^4+^/*β*-glucan solutions of were put into glass bottles with plastic caps and irradiated in a Gamma Co-60 Chamber model GC5000 (BRIT, Munbai, India) at a dose range of 4–14 kGy and dose rate of 2.5–10.0 kGy h^−1^ for the synthesis of SeNPs/*β*-glucan solutions.

### 2.3. Preparation of SeNPs/β-Glucan Powder

The SeNPs/*β*-glucan solution from 80 ppm Se^4+^ in 2% *β*-glucan sample was used to prepare SeNPs powder by spray drying on spray dryer model B290 (Buchi, St. Gallen, Switzerland) and by freeze drying in freeze dryer model GAMMA 1-16 LSCplus (Christ, Lower Saxony, Germany). The SeNPs powder was also prepared by coagulation. For coagulation of SeNPs/*β*-glucan, ethanol (9 volumes) was mixed with 1 volume of SeNPs/*β*-glucan solution. The resulting SeNPs/*β*-glucan coagulate was filtered and washed several times with ethanol and dried in a forced air oven at 60 °C.

### 2.4. Analysis of Se^4+^ in Irradiated Samples

To determine the remain content of Se^4+^ ion in Se^4+^/*β*-glucan solution after irradiation, irradiated Se^4+^/*β*-glucan solutions were centrifuged at 111,400× *g* for 30 min in a model Optima MAX-XP ultracentrifuge (Beckman Coulter, Brea, CA, USA) for separation of SeNPs. The content of Se^4+^ ion in the supernatant was analyzed by a spectrophotometric method using Azure B as a chromogenic regent as described by Mathew and Narayana [34].

### 2.5. Characterization of SeNPs/β-Glucan

The particle size, size distribution and zeta potential of SeNPs solutions were characterized on a Malvern Zetasizer model ZEN5600 (Malvern, Worcestershire, UK) using the Zetasizer Softwave V.7.12. The size and size distribution of the SeNPs/*β*-glucan were also further determined from TEM images obtained on a model JEM1010 transmission electron microscope (TEM, JEOL, Tokyo, Japan) and statistically calculated from about 300 particles, typically for the 80 ppm SeNPs sample synthesized at 8 kGy and dose rate of 10 kGy h^−1^.

The crystalline structure of SeNPs was characterized using an X’Pert PRO–powder diffractometer (Malvern) with parafocusing Bragg–Brentano geometry using CuKα radiation (λθ = 1.5418 Å, U = 40 kV, I = 30 mA). SeNPs/*β*-glucan sample was prepared by centrifuging the sample at 111,400× *g* for 30 min at 10 °C. The sample was dried at room temperature to get the powder form for analysis. Data were scanned with the ultrafast detector X’Celerator or with a scintillator detector equipped with a secondary curved monochromator over the angular range 5–60° (2θ) with 0.02° (2θ) and 0.3 s step^−1^. The software package HighScore Plus was used for data analysis.

Fourier transform infrared (FTIR) spectroscopy analysis was performed to the reveal functional groups involved the synthesis of *β*-glucan and SeNPs/*β*-glucan. The spectra were recorded in the region between 4000 and 400 cm^−1^ with 128 scans at a resolution of 4 cm^−1^ using a FTIR-8100A spectrophotometer (Shimadzu, Kyoto, Japan) linked with a Shimadzu DR-8030 computer system. Samples were prepared in a KBr pellet formed by well-dried mixtures of freeze-dried samples (*β*-glucan and 80ppm-SeNPs/*β*-glucan) and 100 mg of KBr.

The stability of SeNPs/*β*-glucan solution stored at various temperature conditions within 60 days and the change of particle size and size distribution in powder products were evaluated by dynamic light scattering (DLS).

### 2.6. Animal Experimental Design

Fifty-four 12-week-old male BALB/c mice with body weight about 25–30 g were raised in cages at the standard room of Stem Cell Institute followed the guidelines of the Association for Assessment and Accreditation of Animal Care International (AAALAC). After two weeks being raised at 22 °C and 12-h light-dark cycle, mice were divided into 2 groups. A group of 9 mice was used for normal control (N-Ctrl). Another group of 45 mice were subjected to immunosuppression by intraperitoneal injection of CTX with a dose of 100 mg kg^−1^ body weight for 3 consecutive days [35,36]. Then, the CTX-treated mice were randomly divided into five groups (nine mice per group) for experiment as follows: mice in the control group (CTX-Ctrl) were received only distilled water, while mice in other four groups were orally supplied with 0 (only *β*-glucan without SeNPs), 2, 4 and 6 mg SeNPs/*β*-glucan product daily per a kg body weight for 14 days. After supplying with SeNPs/*β*-glucan for 14 days, mice in each group were killed through a cervical dislocation method and used for analysis.

### 2.7. Spleen Index Determination

The spleen was dissected and weighed. The spleen index was calculated by the following equation [25]: Spleen index (mg 10 g^−1^) = [Spleen weight (mg)] × [Animal body weight (g)]^−1^ × 10.

### 2.8. Analysis of Cytokine and Immunoglobulin Indexes in Serum and Spleen

For analysis of cytokine and immunoglobulin indexes in serum, all blood of tested mice in each group was collected and centrifuged at 1438× *g* for 10 min to collect serum. The immune parameters including IgG, IgM, IL-2, IFN-γ and TNF-α in collected serum were determined by suitable ELISA Kits.

For spleen samples, all spleens of each tested mice were collected and conducted according to the method described by Han et al. [36]. Briefly, the collected spleens were soaked with a 70% ethanol solution within 3 min for sterilization and then washed with Hank’s buffer solution twice under sterile condition. A portion of 100 mg of spleen tissue was homogenized in 2 mL physiological saline solution at 4 °C before centrifuging at 1438× *g* for 15 min to collect the supernatant. The concentration of IL-2, IFN-γ, TNF-α, IgG and IgM in spleen was determined in the same way as in serum above.

### 2.9. Determination of Cellular Immunity

The ratios of hematopoietic progenitor cells (CD34-ab81289 for primary antibody and ab150077 for secondary antibody), T-CD4 + cells (CD4-ab269349 antibody conjugated with fluorescein isothiocyanate), B cells (CD20-ab64088 for primary antibody and ab150077 for secondary antibody), natural killer (NK) cells (CD161-ab137059 for primary antibody and ab150077 for secondary antibody), neutrophils (clone 7/4 recognizes the Ly-6B.2 antigen in 129J-ab53453 antibody conjugated with fluorescein isothiocyanate), WBC and lymphocyte which were gated base on two optical detectors: forward scatter (FSC) and side scatter (SSC) in peripheral blood and bone marrow of tested mice were determined using a model FACS ARIA III flow cytometer (BD Science, Franklin Lakes, NJ, USA).

For peripheral blood samples, all red blood cells in blood were lysed by a red blood cell lysis buffer and then washed once with 45 mL Dulbecco’s PBS in 2% fetal bovine serum (FBS). The supernatant was aspirated, and pellets were incubated in a 1% bovine serum albumin (BSA) blocking solution before resuspending in phosphate buffered saline (PBS). Suspended cells were stained for 30 min with the antibodies, washed twice with PBS, and recovered via centrifugation. Flow cytometry was performed using a BD FACSAria™ III cell sorter.

In case of bone marrow samples, femoral segments from shin bones of tested mice were collected for use. The bone segments were cut and then the cell culture medium (DMEM/F12 supplemented with 10% FBS and 1% penicillin-streptomycin) was pumped into the bone marrow three times by a 1 mL-needle. All the cell suspension was transferred to a clean centrifuge tube and centrifuge at 734× *g* for 10 min to obtain bone marrow and treat in the way of peripheral blood sample before analyzing by flow cytometry.

All tests on the mice were conducted according to the ethical guidelines for animal research of (AAALAC) and approved by the Scientific Council of the Biotechnology Center of Ho Chi Minh City (approval No.: 380/QĐ-CNSH). All experiments were repeated three times. Data were statistically analyzed using the ANOVA test. The least significant difference (LSD) at a 5% probability level was used to compare the mean values and the standard deviations were also calculated.

## 3. Results and Discussion

### 3.1. Characteristics of Radiation Synthesized SeNPs

SeNPs have been already synthesized by γ-irradiation using oligochitosan as a stabilizer [16]. In this study, solutions containing various Se^4+^ concentrations in water-soluble *β*-glucan with Mw~25 kDa were irradiated for the preparation of SeNPs. It can be seen from Figure 1 that the doses required for complete reduction of Se^4+^ into Se^0^ (the saturation dose for complete reduction of Se^4+^) in the solutions containing 40, 60, 80, 100 and 120 ppm Se^4+^ were found to be 4, 6, 8, 10 and 12 kGy, respectively. This can be visualized by the color change from light yellow to orange-red of SeNPs/*β-*glucan solutions after irradiation (see Figure 1).

Several papers have reported that Se^4+^ concentration was one of the major parameters affecting the particle size and size distribution of SeNPs products [15,16]. The influence of Se^4+^ concentration on the particle size and size distribution of SeNPs/*β*-glucan synthesized by the γ-irradiation method is presented in Figure 2. The results of a dynamic light scattering (DLS) analysis clearly showed that the particle size of SeNPs increased with the increase of concentration of Se^4+^ in the irradiated samples. Particularly, particle sizes of SeNPs ranging from 64.8 to 110.1 nm were determined in irradiated samples containing 40–120 ppm Se^4+^, respectively. These results were in good agreement with those reported by Jia el al. [15]. In addition, the particle size distribution found to display a Gaussian distribution in all products. Among them, the product obtained from the irradiated sample containing 80 ppm Se^4+^ showed the narrowest distribution.

On the other hand, the results in Figure 3 also show the effect of the pH on the particle size and its distribution. It can be clearly seen that the average particle size of SeNPs in the products was not significantly reduced, but the size distribution was greatly increased by an increase in the pH level. The reason may be due to an increase of the swelling of *β*-glucan molecules in solution with higher pH, leading to a looser network structure of the stabilizer. Jia et al. [15] reported that lentinan (branched *β*-(1,3)-D-glucan isolated from the fruiting bodies of *Lentinus edode*) played an important role in the dispersion of SeNPs in solution.

The dose rate has been reported to have significant effect on the particle size of nano-colloidal products synthesized by irradiation [37,38]. Hien et al. [37] reported that an increase of dose rate from 0.5 to 5 kGy h^−1^ led to a corresponding decrease of the size of gold nanoparticles (AuNPs) from 9.5 to 5 nm, respectively. Khoa et al. [38] informed that a higher dose rate led to a narrower size distribution of AuNPs. In the present study, it can be seen from Figure 4 that the particle size of SeNPs was decreased by the increase of dose rate. In addition, the size distribution of SeNPs also became narrower when the sample was irradiated with a higher dose rate. The reason for this phenomenon is the competition between the adsorption of Se^4+^ onto the resultant selenium clusters and the reduction reaction of Se^4+^ → Se^3+^ → Se^2+^ → Se^1+^ → Se^0^ to form new clusters. At a high dose rate, the reduction reaction is predominant and therefore there are many new clusters leading to smaller SeNPs formed. On the contrary, at a low dose rate the adsorption of Se^4+^ onto clusters is predominant; therefore, SeNPs will be larger.

The above results suggested that the SeNPs/*β*-glucan synthesized by irradiation of solution containing 80 ppm Se^4+^ in pH~5 at 8 kGy with dose rate of 10 kGy h^−1^ is the suitable product selected for further characterization and investigation in mice as well. Results from Figure 5c indicate that the particle size of SePNs in this product measured by TEM was about 65.3 nm and smaller than that estimated by DLS. According to Souza, et al. [39], the DLS mean size was approximately 20% higher than the TEM mean size because DLS gives a hydrodynamic size, which is the size of the nanoparticle plus the liquid layer around the particles, but the size measured by TEM represents the actual size of the nanoparticle. In addition, the zeta potential of this product was about −6.9 mV (Figure 5b). This factor is one of the key particle properties that can affect the particle stability as well as its cell adhesion. These results are in good agreement with those obtained for SeNPs synthesized by a chemical method using *β*-glucan isolated from the fruiting bodies of *Lentinus edodes* as stabilizer [15].

Figure 6 shows the FTIR spectra of SeNPs/*β-*glucan prepared from the solution containing 80 ppm Se^4+^ and 2% water-soluble *β-*glucan and water-soluble *β-*glucan. It can be seen from Figure 6 that all typical absorption peaks of *β-*glucan and SeNPs/*β-*glucan including –OH (3383 cm^−1^), –CH_2_ (2986 cm^−1^), –C=O (1640 cm^−1^), C–O–C (1156 cm^−1^) and *β*-D-glucan (890 cm^−1^) are present, however, the typical peaks assigned for –OH and –C=O were significantly shifted from 3383 cm^−1^ to 3390 cm^−1^ and from 1640 cm^−1^ to 1647 cm^−1^ in the spectrum of SeNPs/*β-*glucan, respectively. The changes of –OH group and C=O linkage vibrations of *β-*glucan in SeNPs/*β-*glucan spectrum may be attributed to the hydrogen bond-like interactions between *β-*glucan molecules and SeNPs and indicative of a complexation between selenium and *β-*glucan molecules in the product via the steric linkage (Se…O). Our results are in good agreement with those of SeNPs/dextran reported by Hien et al. [7] and SeNPs/oligochitosan reported by Duy et al. [16].

The crystal structure of SeNPs after irradiation was also confirmed by X-ray diffraction (XRD) analysis. The typical XRD pattern in Figure 7 indicated that SeNPs exhibited diffraction peaks at 23.7° (100), 29.5° (101), 42.7° (110), 44.8° (102), 46.2° (111), 51° (201) and 65.3° (210) attributed to the crystal structure of SeNPs and these results are in good agreement with those reported in previous papers [15,16,40,41]. Based on the above results, we can conclude that a high purity and stable SeNPs/*β-*glucan was successfully synthesized by irradiation, and the capped mechanism illustrated in Figure 8 can be proposed.

The stability of a colloidal nanoparticle solution is the most important for application and it depends on several factors including concentration, type of stabilizer, pH of solution and temperature during storage [7,42]. The properties of SeNPs/*β-*glucan stored for 60 days at 0, 4 and 25 °C in this study are presented in Figure 9. It can be clearly seen that the particle size of storage SeNPs/*β-*glucan slightly increased under 0 °C storage conditions, but these characteristics increased significantly in samples stored at 4 °C and increased strongly in samples kept at room temperature (25 °C). In addition, the particle size distribution was rather larger at higher storage temperature. Zeta potential is an important indicator of the stability of nanocolloidal suspensions. Figure 5b indicates that the zeta potential of SeNPs/*β-*glucan was about −6.9 mV (approximately neutral); therefore, the temperature strongly affected the stability of this solution during storage. According to Bai et al. [25], nanoparticles are easily aggregated to form larger ones at high temperature due to Brownian movement. Duy et al. [16] also informed that the SeNP size in SeNPs/oligochitosan stored at 27 °C increased more rapidly than that at 4 °C. The authors also reported that SeNPs/oligochitosan became unstable after storage for more than 28 days and turned into a black bulk mass after 42 days of storage at 25 °C. Thus, it is challenging to maintain a colloidal SeNPs solution for a long time at normal temperature and the change of SeNPs solution into powder form is a suitable way for improving the applications of this product. Thao et al. [43] prepared AuNPs/dextran by spray drying, coagulation and centrifugation methods. These authors mentioned that the change of particle size in AuNPs powders prepared by the spray drying and coagulation techniques was less than that in AuNPs made by centrifugation. Recently, Duy et al. [16] reported that the particle size (estimated by TEM images) of SeNP/oligochitosan solution increased slightly from 41.8 nm to 43.8 nm in the powder form obtained after dry spraying. In the present study, the DLS analysis results (Figure 10) indicate that particle size and size distribution of SeNPs/*β-*glucan solution after changing to the powder form were increased. Particularly, the particle size of SeNPs/*β-*glucan increased slightly from 92 nm in the original solution to 95, 117.6 and 132.3 nm in powder products prepared by freeze drying method, coagulate and spray draying techniques, respectively. From the above results, the powder sample made from SeNPs/*β-*glucan solution contained 80 ppm Se^4+^ and 2% *β-*glucan was selected for further evaluation of the immunostimulant activity in mice.

### 3.2. Effect of SeNPs/β-Glucan on Immune Parameters in Peripheral Blood

According to Ahlmann and Hempel [44], CTX is one of the most commonly used chemotherapy drug for various tumors, cancers and marrow transplantation. Therefore, a CTX-induced immunosuppressed mice model is usually used to evaluate the immunoregulatory activity of immunostimulant compounds [45]. In the present study, this chemical was used to induce immunosuppression in tested mice. It can be seen that all of immune indexes were strongly reduced in CTX-induced immunosuppressed mice and the oral administration of SeNPs/*β-*glucan displayed a strong ameliorating effect on these deficiencies.

In peripheral blood, white blood cells (WBCs) are a very important part of organisms’ way of resisting external microorganisms and the differentiation in WBCs also affects the immune responses. The effects of SeNPs/*β-*glucan supplied at various concentrations on cellular immune parameters in the peripheral blood of tested mice are displayed in Figure 11 and Appendix A. It can be seen that the oral administration of SeNPs/*β-*glucan at a concentration of 2–6 mg kg^−1^ body weight enhanced WBCs in peripheral blood of CTX-induced immunosuppressed mice. 

Compared to the content of WBC of peripheral blood in CTX-control mice (20.8%), the supplementation of 4–6 mg kg^−1^ enhanced the contents of these immune cells to 35–45.1%. In addition, the results on differentiated WBCs also showed that the ratio of neutrophils (from 0.27% to 1.87–2.73%), lymphocytes (from 11.6% to 21.7–25.2%), B cells (from 0.15% to 1.02–1.57%) CD4+ cells (from 0.47% to 1.97–5.57%) and NK cells (from 1.07% to 2.57–3.53%) also significantly increased in peripheral blood of mice in group supplemented with 4–6 mg SeNPs/*β-*glucan.

Generally, immune molecules including immunoglobulins are the main factors in the immune system of the body besides immune cells and organs. The results in Figure 12 also demonstrate that compared to IgG (687.3 µg mL^−1^), IgM (137.1 µg mL^−1^), TNF-α (35.5 µg mL^−1^) and IFN-γ (1.2 µg mL^−1^) in the CTX-control group, the supplementations of 2–6 mg SeNPs kg^−1^ increased these levels to 243– 285.7, 1491–1730.1, 54.6–62.4 and 1.6–2.11 µg mL^−1^, respectively. The highest levels of these indexes were found in the group fed with 6 mg SeNPs kg^−1^ and were almost the same as those of normal control animals.

Although the ameliorating effect of SeNPs on chemical-induced immunosuppression in mice has not been reported, Menon et al. [17] and Gao et al. [46] have indicated that SeNPs capped by glutathione and BSA with particle size about 22.7 nm could enhance the chemotherapeutic activity by acting as a functional division of redox center and protecting tissues from cellular damage caused by ROS. Shakibaie et al. [47] indicated that the total number of WBCs in normal mice was significantly increased by feeding with SeNPs suspended in deionized water with particle size in range of 80–220 nm at a concentration of 20 mg kg^−1^. SeNPs was also reported to restore WBCs in radiation-induced immunosuppressed mice. Yazdi et al. [48] also demonstrated that the daily administration of biogenic SeNPs suspended in deionized water with particle size in range of 80–220 nm at a dose of 100 µg per mouse led to an increase in the WBCs including total number of neutrophil and lymphocyte cells in blood of X-ray irradiated mice. Duy el at. [16] have reported that the daily administration of 20 µg SeNPs/oligochitosan with particle size about 43.8 nm synthesized by γ-irradiation also showed a strong effect on WBC recovery in the blood of γ-ray irradiated mice. Recently, Raahati et al. [49] also demonstrated that the oral supplementation of SeNPs capped by Tween 20 with particle size about 48 nm at a dose of 100 µg per head led to a significant increase of *V. cholera*-specific immunoglobulins (IgG and IgA) and interleukin (IL-4 and IL-5) responses in the serum of mice challenged with *V. cholera* bacteria. The above information indicates that the immunostimulant activity of SeNPs seems to be not affected by capping polymers. It can be seen clearly from Figure 11 and Figure 12 that both cellular immune factors (WBCs, neutrophil, lymphocyte, B cells, CD4+ cells and NK cells) and humoral immune indexes (IgG, IgM, TNF-α and IFN-γ) in peripheral blood CTX-control and the *β-*glucan treated mice were not significantly different. Although *β-*glucan did not show a significant effect on the immunostimulation in CTX-induced immunosuppressed mice, this natural polymer has been widely applied in the food and pharmaceutical industries due to its excellent biocompatibility [27,28,29]. In addition, Zhang et al. [50] also demonstrated that SeNPs with particle sizes ranging from 5 to 200 nm did not show a size-dependent effect in upregulating seleno-enzymes, both in cultured cells and mice. The results in the present research reveal that the SeNPs/*β-*glucan prepared by γ-irradiation significantly ameliorated the differentiated WBCs, level of immunoglobulins (IgG and IgM) and cytokines (TNF-α and IFN-γ) in peripheral blood of CTX-induced immunosuppressed mice.

### 3.3. Effect of SeNPs/β-Glucan on Immune Parameters in Bone Marrow

In mammals and humans, bone marrow is a hematopoietic organ and central immune organ of humans and mammals and bone marrow suppression is one of the most obvious side effects of drug chemotherapy, including CTX treatment [36,44]. It was reported that the lifetime of WBCs in peripheral blood is short and they need to be continuously complemented from bone marrow [51]. So far, the effect of SeNPs on cell differentiation or cytokine levels of bone marrow suspension in mice immunosuppressed by chemicals has not been reported. Therefore, effect of SeNPs/*β-*glucan on cytokines of bone marrow suspension of CTX-treated mice was evaluated in this study. The analysis results in bone marrow of tested mice from our investigation indicated that the ratio of WBC was significantly increased in bone marrow suspensions of mice in all groups administrated with SeNPs/*β-*glucan compared to the CTX-control group supplied with only distilled water (Figure 13 and Appendix A). Particularly, the WBC ratio in bone marrow suspension of the CTX-control group was determined at 11.2% and these ratios in groups fed with 2, 4 and 6 mg SeNPs/*β-*glucan were significantly increased to 52.3, 75.5 and 84.6%, respectively. On the other hand, the oral supplementation with 6 mg SeNPs product also led to the strongest effects on the increase of lymphocytes (44.4%) and CD34+ cells (3.6%) in tested mice. The increasing proliferation of immune cells in bone marrow suspension could activate the immune system and modulate the immune activity in immunosuppressed mice. These results also revealed clearly that SeNPs/*β-*glucan prepared by γ-irradiation significantly ameliorated immune cell proliferation in bone marrow, which is very important for a continuously maintenance at a high level of cellular and humoral immune factors in the peripheral blood of immunosuppressed mice. The suitable concentration for oral administration of this product was determined to be about 4–6 mg kg^−1^.

### 3.4. Effect of SeNPs/β-Glucan on Immune Parameters in Spleen

Beside bone marrow, the spleen is also an important immune organ due to its close relation to cellular immunity and humoral immunity. Raahati et al. [49] and Yazdi et al. [13] have reported that biogenic SeNPs with a concentration of 100 µg per day significantly enhanced the levels of cellular immunomodulatory components (granzyme B, IL-12, IFN-γ, and IL-2) in the spleen cells of tumor-bearing mice. The results in Figure 14 indicate that the spleen index of CTX-control group mice was 16.3 mg 10 g^−1^ and this value was increased in to 21.5–37.3 mg 10 g^−1^ in groups fed with SeNPs/*β-*glucan at concentrations of 2–6 mg kg^−1^, respectively. In addition, Figure 14 also shows that the levels of IgM, IgG, TNF-α, IFN-γ and IL-2 in the CTX-control group were found to be about 5.0, 8.7, 23.4, 1.2 and 0.78 µg mL^−1^, respectively. These levels were much lower than those in the normal control group and they were strongly recovered in groups supplemented with SeNPs/*β-*glucan. Particularly, the supplementation with 2–6 mg kg^−1^ SeNPs/*β-*glucan increased IgM, IgG, TNF-α, IFN-γ and IL-2 levels to 12.2–17.3, 28.2–37.8, 56.2–97.2, 2.0–2.9 and 2.7–4.6 µg mL^−1^, respectively. The highest levels of these indexes were found in the group of mice fed with 6 mg kg^−1^, being almost the same as those of the normal control animals.

The above results in the tested mice clearly reveal that the oral administration of SeNPs/*β-*glucan synthesized by γ-ray irradiation at a daily dose of 6 mg kg^−1^ significantly ameliorated the immune system suppression caused by CXT treatment in mice.

## 4. Conclusions

SeNPs with diameter from 64.8 to 110.1 nm capped by water soluble *β-*glucan were synthesized by gamma Co-60 ray irradiation. The daily supplementation of radiation-synthesized SeNPs/*β-*glucan at 6 mg kg^−1^ strongly restored the cellular immune factors (WBC, neutrophils, monocytes, lymphocytes, B cells and NK cells) and humoral immune indexes (IgM, IgG, TNF-α, IFN-γ and IL-2) in peripheral blood, bone marrow and spleen of CTX-induced immunosuppressed mice. Due to the excellent biocompatibility of *β-*glucan, as well as the unique attributes of SeNPs, the SeNPs/*β-*glucan product may be a candidate for application as a functional food for the prevention of immunosuppression in cancer chemotherapy. The in vivo release in blood and biodistribution of SeNPs will be further evaluated to elucidate the pathway of its action.

## Figures and Tables

**Figure 1 nanomaterials-11-02439-f001:**
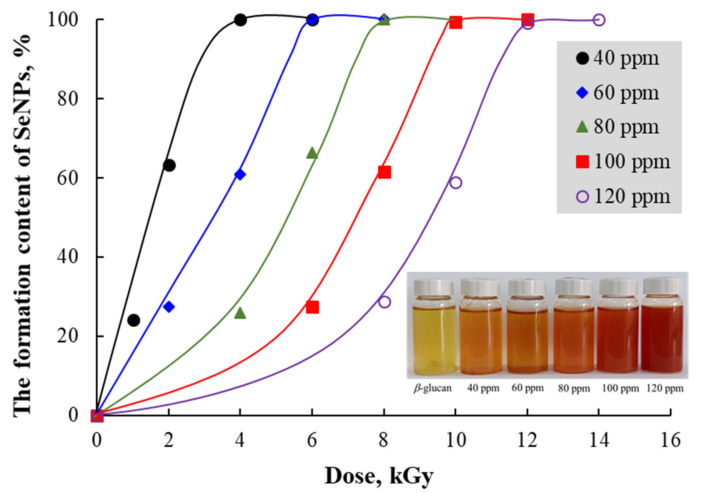
The saturated dose for radiation synthesis of SeNPs/*β*-glucan samples with various Se^4+^ concentration.

**Figure 2 nanomaterials-11-02439-f002:**
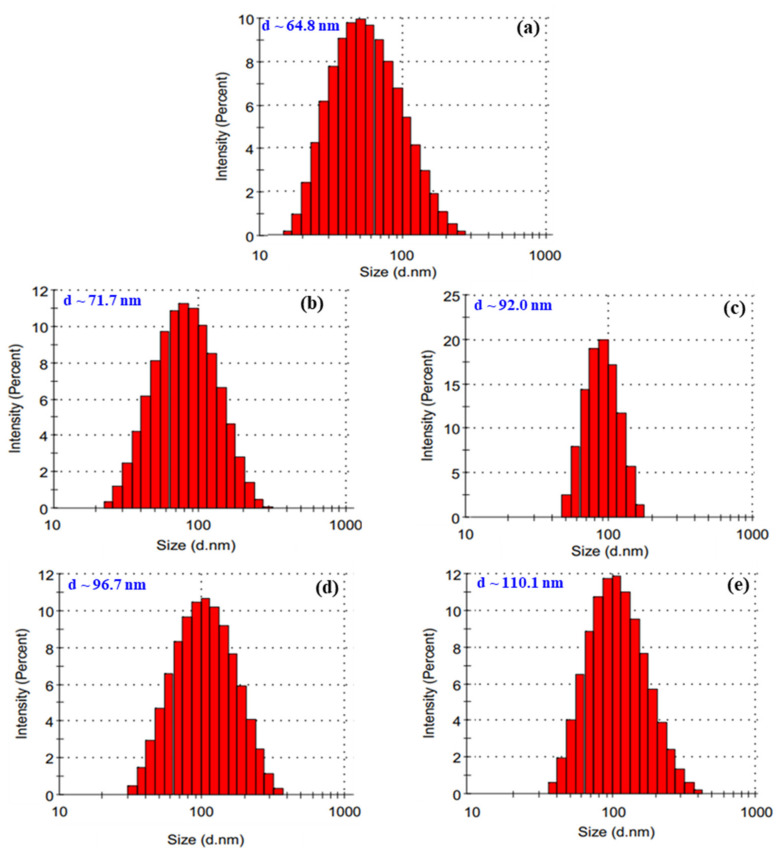
Particle size and size distribution of SeNPs/*β*-glucan samples with various Se^4+^ concentrations syntheiszed by irradiation. (**a**) 40 ppm; (**b**) 60 ppm; (**c**) 80 ppm; (**d**) 100 ppm and (**e**) 120 ppm.

**Figure 3 nanomaterials-11-02439-f003:**
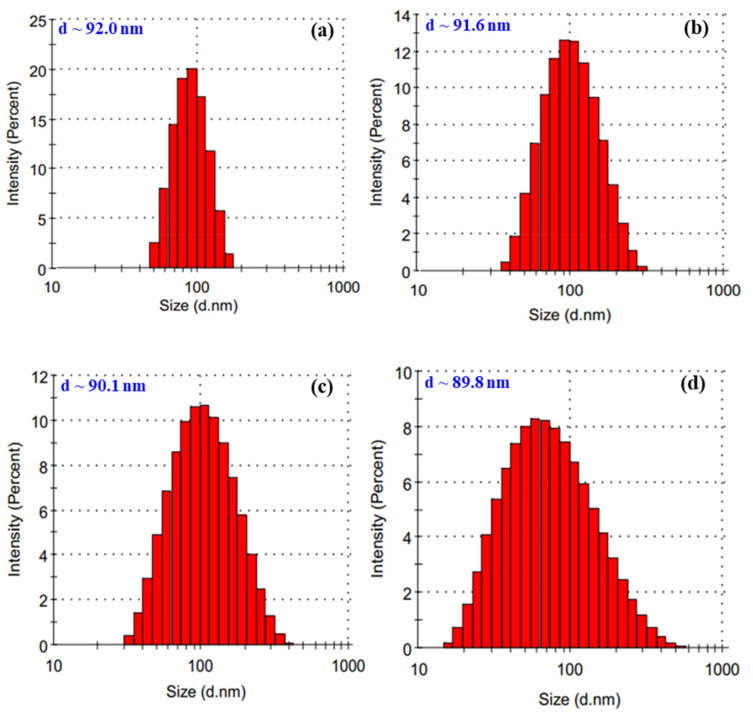
Particle size and size distribution of SeNPs/*β*-glucan samples of 80 ppm SeNPs synthesized in various pH conditions. (**a**) pH~5; (**b**) pH~6; (**c**) pH~7 and (**d**) pH~8.

**Figure 4 nanomaterials-11-02439-f004:**
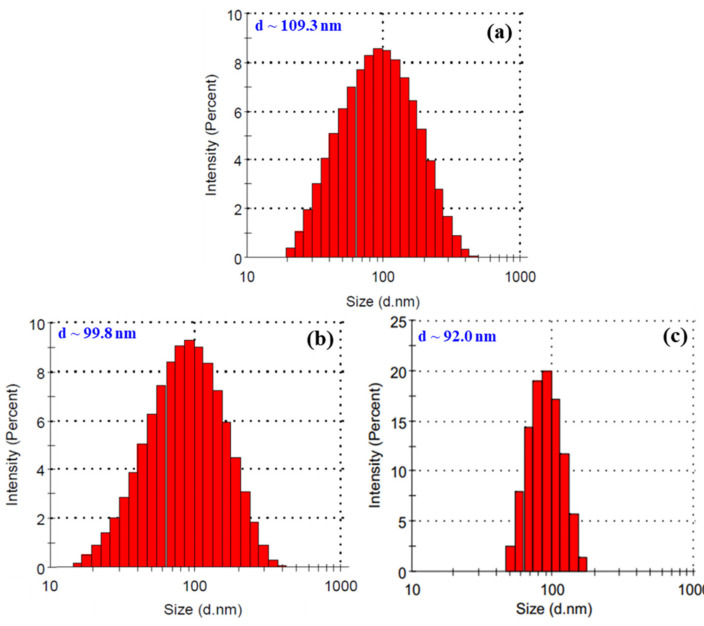
Particle size and size distribution of SeNPs/*β*-glucan samples synthesized at dose rates of 2.5 (**a**), 5 (**b**) and 10 kGy h^−1^ (**c**).

**Figure 5 nanomaterials-11-02439-f005:**
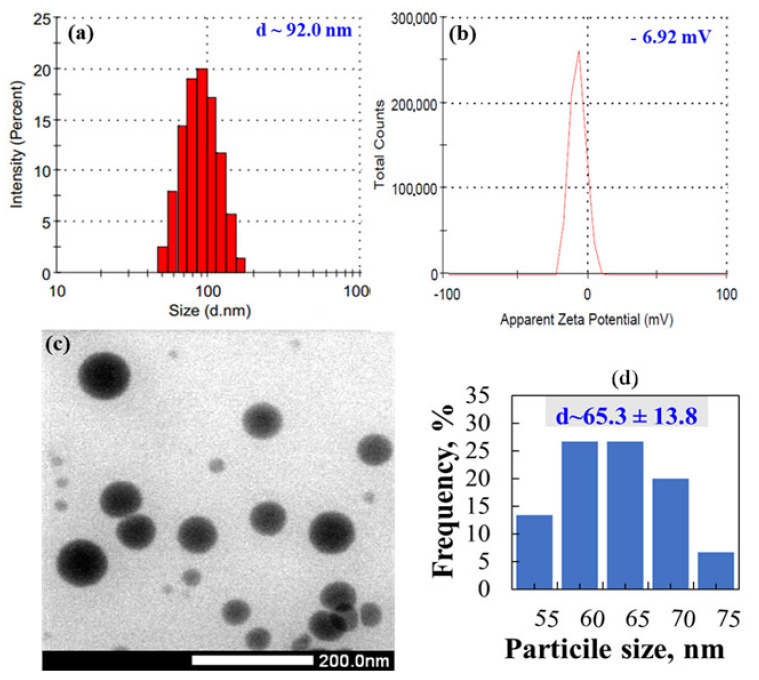
Particle size, size distribution (**a**) and zeta potential (**b**) TEM image (**c**) and size distribution histogram (**d**) of SeNPs/*β*-glucan synthesized by gamma irradiation at dose of 8 kGy and dose rate of 10 kGy h^−1^.

**Figure 6 nanomaterials-11-02439-f006:**
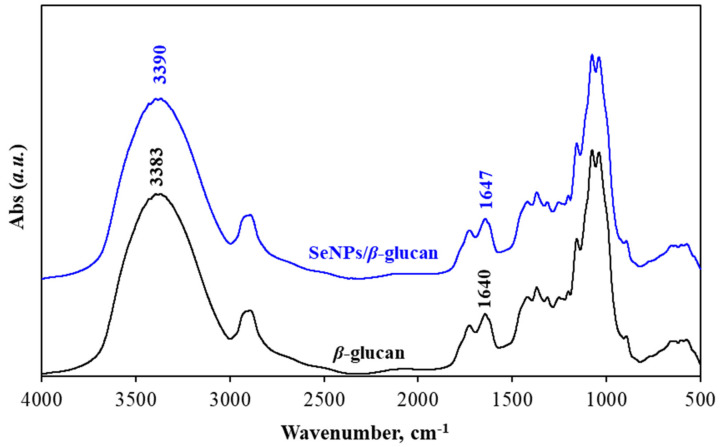
FTIR spectra of *β*-glucan and SeNPs/*β*-glucan synthesized by gamma irradiation at dose of 8 kGy and dose rate of 10 kG h^−1^.

**Figure 7 nanomaterials-11-02439-f007:**
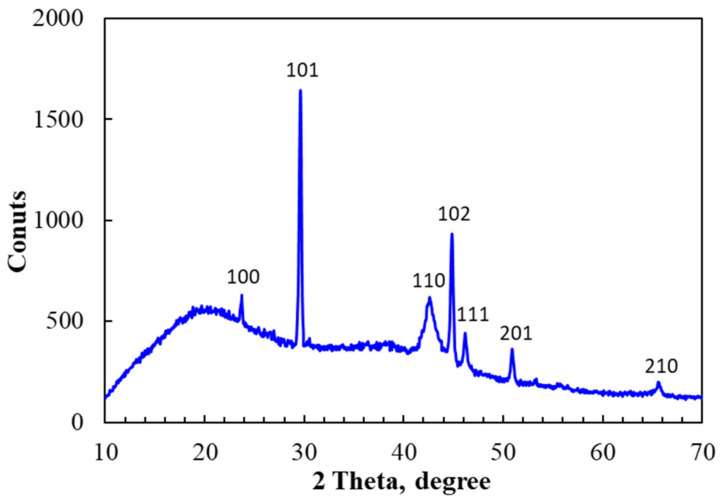
XRD pattern of SeNPs/β-glucan synthesized by gamma irradiation at a dose of 8 kGy and dose rate of 10 kG/h.

**Figure 8 nanomaterials-11-02439-f008:**
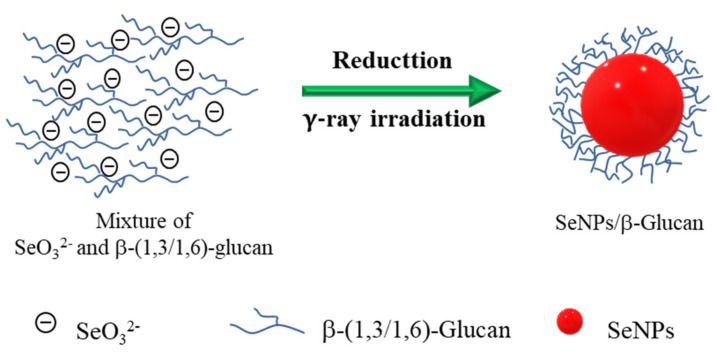
Schematic representation of the synthesis of SeNPs capped by β-glucan using gamma irradiation for reducing Se^4+.^3.2. Stability of the SeNP/β-glucan solution by storage.

**Figure 9 nanomaterials-11-02439-f009:**
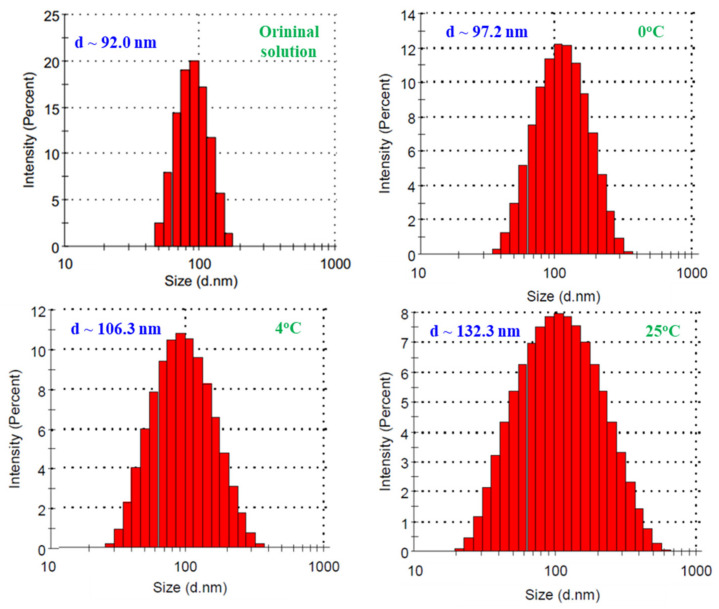
Particle size and size distribution of SeNPs/*β*-glucan solution after storage for 60 days at various temperatures.

**Figure 10 nanomaterials-11-02439-f010:**
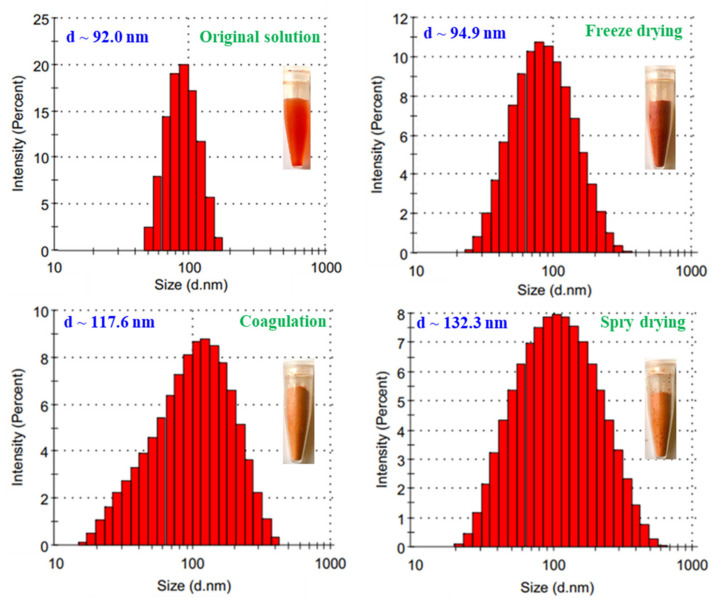
Particle size and size distribution of SeNPs/*β*-glucan solution and after changing into powder form by different techniques.

**Figure 11 nanomaterials-11-02439-f011:**
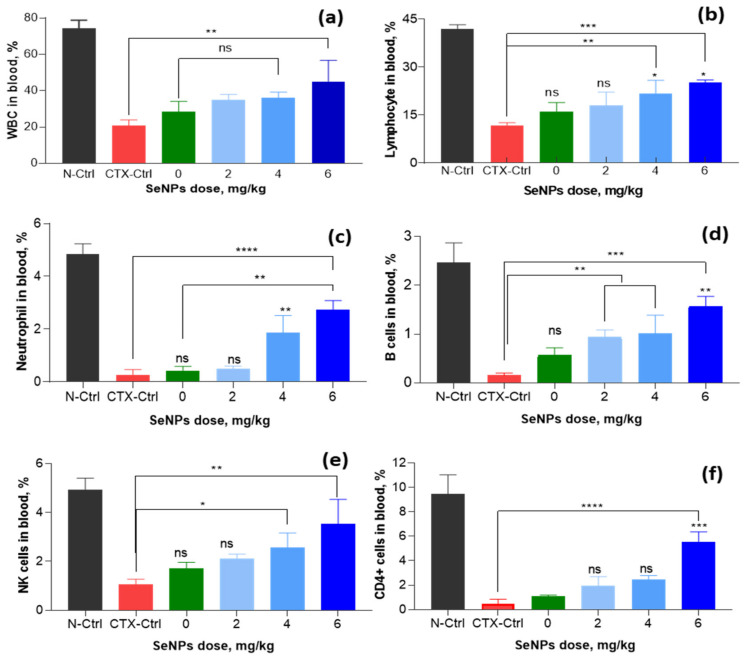
Effect of SeNPs/*β*-glucan concentration on cellular immunity indexes in peripheral bloods of CTX-induced immunosuppressed mice. (**a**–**f**) are WBCs, lymphocytes, neutrophils, B cells, NK cells and CD4+ cells, respectively; N-Ctrl: Normal control mice received only distilled water; CTX-Ctrl: CTX-induced immunosuppressive mice received only distilled water; in supplemented groups, mice were orally supplied with 0 (only β-glucan without SeNPs), 2, 4 and 6 mg SeNPs kg^−1^. Significant differences were compared with the CTX-Ctrl. ns, not significantly different at *p* > 0.05; *, significant different at *p* < 0.05; **, significant different at *p* = 0.01, *** significant different at *p* < 0.01; ****, significant different at *p* < 0.001. Data were statistically analyzed using the ANOVA test and expressed as means ± SD, *n* = 3.

**Figure 12 nanomaterials-11-02439-f012:**
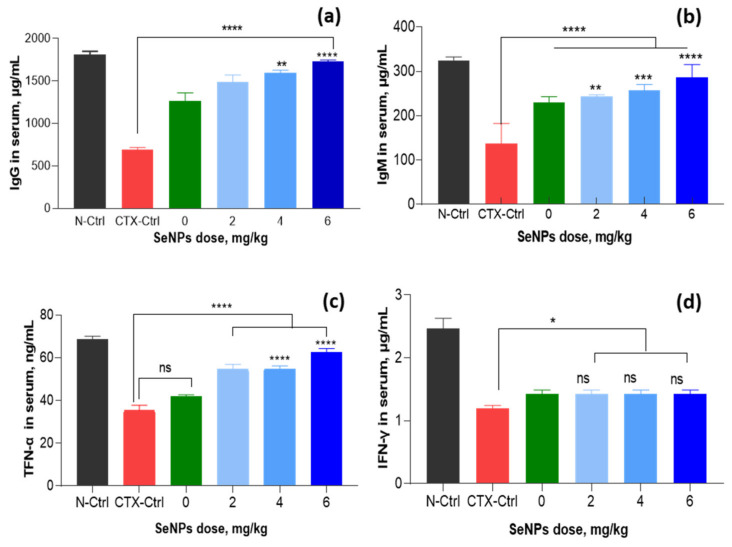
Effect of SeNPs/*β*-glucan concentration on humoral immunity factors in serum of CTX-induced immunosuppressed mice. (**a**–**d**) are IgG, IgM, TFN-α and IFN-γ indexes, respectively; N-Ctrl: Normal control mice received only distilled water; CTX-Ctrl: CTX-induced immunosuppressive mice received only distilled water; in supplemented groups, mice were orally supplied with 0 (only β-glucan without SeNPs), 2, 4 and 6 mg SeNPs kg^−1^. Significant differences were compared with the CTX-Ctrl. ns, not significantly different at *p* > 0.05; *, significant different at *p* < 0.05; **, significant different at *p* = 0.01, *** significant different at *p* < 0.01; ****, significant different at *p* < 0.001. Data were statistically analyzed using the ANOVA test and expressed as means ± SD, *n* = 3.

**Figure 13 nanomaterials-11-02439-f013:**
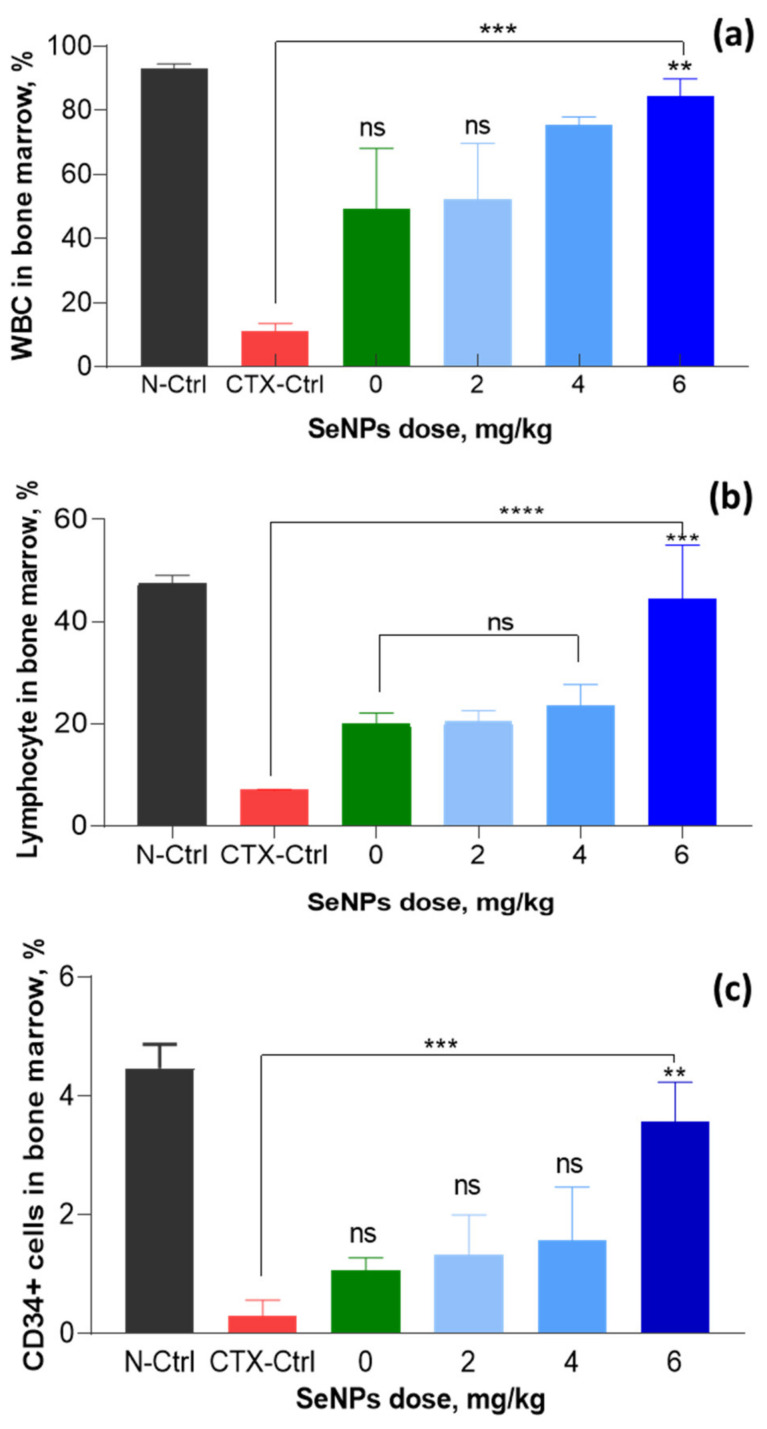
Effect of SeNPs/*β*-glucan concentration on cellular immunity indexes in bone marrow of CTX-induced immunosuppressed mice. (**a**–**c**) are WBCs, lymphocytes and CD34+ cells, respectively; N-Ctrl: Normal control mice received only distilled water; CTX-Ctrl: CTX-induced immunosuppressed mice received only distilled water; in supplemented groups, mice were orally supplied with 0 (only *β*-glucan without SeNPs), 2, 4 and 6 mg SeNPs kg^−1^. Significant differences were compared with the CTX-Ctrl. ns, not significantly different at p > 0.05; **, significant different at *p* = 0.01, *** significant different at *p* < 0.01; ****, significant different at *p* < 0.001. Data were statistically analyzed using the ANOVA test and expressed as means ± SD, *n* = 3.

**Figure 14 nanomaterials-11-02439-f014:**
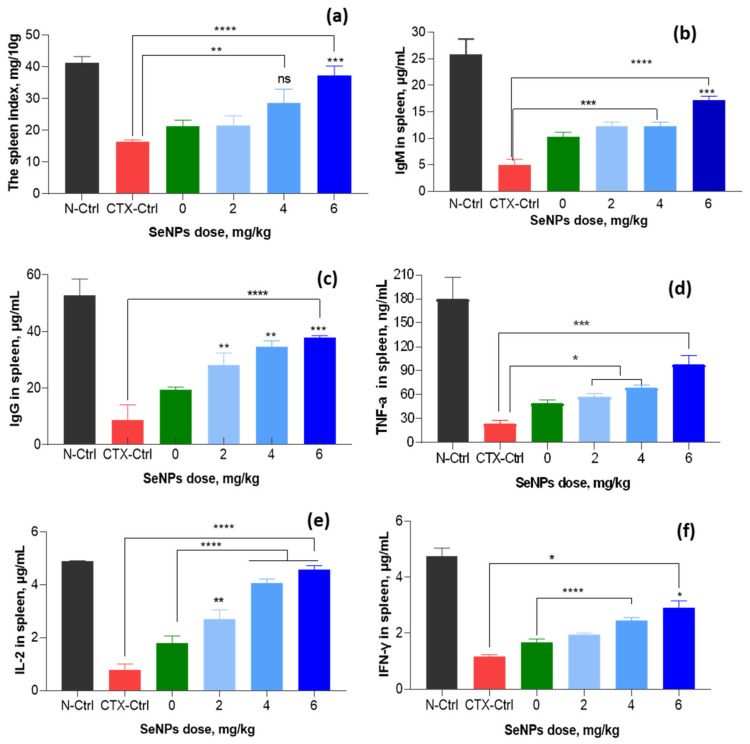
Effect of SeNPs/*β*-glucan concentration on immunity factors in spleen of immunosuppressed mice. (**a**–**f**) are spleen, IgM, IgG, TFN-α, IL-2 and IFN-γ indexes, respectively; N-Ctrl: Normal control mice received only distilled water; CTX-Ctrl: CTX-induced immunosuppressive mice received only distilled water; in supplemented groups, mice were orally supplied with 0 (only β-glucan without SeNPs), 2, 4 and 6 mg SeNPs kg^−1^. Significant differences were compared with the CTX-Ctrl. ns, not significantly different at *p* > 0.05; *, significant different at *p* < 0.05; **, significant different at *p* = 0.01, *** significant different at *p* < 0.01; ****, significant different at *p* < 0.001. Data were statistically analyzed using the ANOVA test and expressed as means ± SD, *n* = 3.

## Data Availability

The data presented in this study are available on a reasonable request from the corresponding author.

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
