# Peer review of "Radiation Synthesis of Selenium Nanoparticles Capped with β-Glucan and Its Immunostimulant Activity in Cytoxan-Induced Immunosuppressed Mice"

_nanomaterials, 2021, doi:10.3390/nano11092439_

Round 1

Reviewer 1 Report

The manuscript “Radiation Synthesis of Selenium Nanoparticles Capped in β-Glucan and Its immunostimulant Activity in Cytoxan-induced Immunosuppression Mice” describes the development of Selenium Nanoparticles and its potentials in immune regulation.  The rational looks sound. However, many aspects still need improvement.

Point 1, The authors need to further clarify the purposes of the study. To my understanding, the authors want to combine the immune regulating effects of SeNPs and b-Glucan. If so, more controls are required to address this issue, such as groups of single SeNPs, groups of Glucan, and groups of SeNPs+Glucan, etc. In another word, what are the advantages of SeNPs/b-Glucan and is it necessary to synthesize the materials?

Point 2, Experimental design issues. Though authors found some immune regulating effects of the NPs, some basic characteristics of the NPs were not clarified. The loading and encapsulation efficiencies of the NPs? The PK of the Se/Glucan? The release of Glucan from the NPs in blood, or at least under a simulated environment in vitro. All these features may help explain the observed findings.

Point 3, More details are required in the methods. Line 102, 111.4g is right? How long? Line 61, show the gating strategies for peripheral blood cells, and this may be added as supplementary data. What biomarkers did the authors use to identify cell populations?

Point 4, Figure issues. Change the X axel to the actual concentrations of Glucan. Change the X axel labelling “concentration” to “dose”.  Supplement more details in the captions. Repeating numbers? Statistical methods? Objectives of the data represent (animals, measurements…).

Point 5, Grammatical issues. Line 57-58. Line 86-87. Line 478, why F was capitalized? Check other similar issues.

Point 6, Can authors draw a schematic graph to show the structure of the NPs?

Reviewer 2 Report

This manuscript optimized the synthesis conditions of selenium nanoparticles capped in β-2 Glucan and evaluate the one of SeNPs/β-glucan in mouse. Although this manuscript is useful and gives us the important information for developing the SeNPs as immunostimulant for oral administration, the point and originality of this study are unclear. The manuscript should undergo revisions for making it suitable for publication.

1) It is necessary to describe in the results and discussion part what are the advantages of SeNPs/β-glucan compared to the other nanoparticles that have been reported.

2) It is also necessary to compare how they function in comparison to other SeNPs nanoparticles.

3) Please describe the effect of size and size distribution of SeNPs/β-glucan on immunostimulant activity in vivo.

4) In the conclusion part, author describe the suitable concentration SeNPs/β-glucan is 6 mg kg-1 body weight. However, author did not use high dose more than 6 mg kg-1. I think the conclusion is overstated with no evidence.

Round 2

Reviewer 1 Report

The concerns have been addressed satisfactorily and the present form is acceptable.